# Backward THz Emission from Two-Color Laser Field-Induced Air Plasma Filament

**DOI:** 10.3390/s23104630

**Published:** 2023-05-10

**Authors:** Yuxuan Chen, Yuhang He, Liyuan Liu, Zhen Tian, Jianming Dai, Xi-Cheng Zhang

**Affiliations:** 1Center for Terahertz Waves and School of Precision Instrument and Opto-Electronics Engineering, Tianjin University, Tianjin 300072, China; cyxququ@tju.edu.cn (Y.C.); 1020202023@tju.edu.cn (Y.H.); lyliuma@tju.edu.cn (L.L.); 2The Institute of Optics, University of Rochester, Rochester, NY 14627, USA; xi-cheng.zhang@rochester.edu

**Keywords:** THz wave generation, two-color laser fields, laser-induced air plasma filament

## Abstract

Two-color laser field-induced plasma filaments are efficient broadband terahertz (THz) sources with intense THz waves emitted mainly in the forward direction, and they have been investigated intensively. However, investigations on the backward emission from such THz sources are rather rare. In this paper, we theoretically and experimentally investigate the backward THz wave radiation from a two-color laser field-induced plasma filament. In theory, a linear dipole array model predicts that the proportion of the backward emitted THz wave decreases with the length of the plasma filament. In our experiment, we obtain the typical waveform and spectrum of the backward THz radiation from a plasma with a length of about 5 mm. The dependence of the peak THz electric field on the pump laser pulse energy indicates that the THz generation processes of the forward and backward THz waves are essentially the same. As the laser pulse energy changes, there is a peak timing shift in the THz waveform, implying a plasma position change caused by the nonlinear-focusing effect. Our demonstration may find applications in THz imaging and remote sensing. This work also contributes to a better understanding of the THz emission process from two-color laser-induced plasma filaments.

## 1. Introduction

Plasma-based THz sources have attracted much attention owing to their extremely high electric field and ultra-broadband emission spectrum [1,2], which are vital for the development of many potential THz applications [3,4]. Different experimental schemes, such as two- or multi-color laser excitation [5,6,7,8,9] and high voltage DC-bias [10,11], were proposed to promote the emission efficiency. A common feature of plasma-based THz sources is the conical emission pattern in the forward direction [12,13,14,15,16], which is related to the interference of the local dipole emitters generated behind the laser ionization front. Previous investigations on the THz angular distribution of plasma-based THz sources mainly focus on the forward hemisphere, where the emitted THz wave is concentrated. However, backward THz emission from single-color induced plasma filaments was theoretically predicted [17,18] and experimentally observed in an argon cluster [19]. Besides, the backward THz emission from two-color laser-induced plasma was theoretically analyzed in comparison with its forward counterpart [20]. Recently, backward THz emission has been experimentally demonstrated using two-color laser-induced micro-plasma [21], which is less affected by the interference due to the sub-millimeter plasma length. Whereas, in practice, the length of plasma filament in the scheme of two-color excitation ranges from a few millimeters to tens of millimeters to obtain an intense THz emission in the forward direction. In this case, the forward THz emission is predominant, and very few investigations on the backward THz emission have been carried out. 

In this article, we systemically investigate the backward THz emission from a two-color laser-induced plasma filament. First, we provide a full angular distribution of THz wave emitted from two-color laser-induced plasma filaments of different lengths calculated by the linear dipole array model. Then, we detect the backward emitted THz wave from a two-color laser-induced plasma filament with a length of ~5 mm through electric-optic sampling (EOS). The spectrum, pump pulse energy dependence, and the laser chirp dependence of the backward THz emission are analyzed. The results show that the spectrum of the backward THz wave is centered at ~0.5 THz with a bandwidth of ~0.7 THz. The dependence of the backward THz amplitude on the laser pulse energy shows essentially the same trend as the forward counterpart. Besides, when the laser pulse energy changes, a temporal shift of the THz wave peak occurs, which can be seen as evidence of the backward emitted THz wave. Our work may help to further understand the THz emission process from two-color laser-induced plasma filament.

## 2. Theoretical and Experimental Basis

Figure 1a shows the schematic diagram of forward and backward THz emission from a two-color laser-induced plasma filament. The two-color laser fields induce electron-ion separation in the plasma, leaving a string of dipoles behind the laser ionization front. Through the interference of the electromagnetic wave emitted from the dipoles, a strong THz wave can be collected in the forward direction. Meanwhile, a much weaker THz wave is emitted in the backward direction from the plasma filament. This THz emission process can be well interpreted by a simplified dipole array model [14]. In this model, the THz wave is emitted from a linear dipole array with a length of *L* in a uniform plasma. THz emission of a single dipole with momentum perpendicular to the laser axis (*z*) can be expressed as:(1)$$dE_{dp}\left(\omega_{THz},\theta ,z\right)\propto \frac{\exp\left[i\Phi \left(\omega_{THz},\theta ,z\right)\right]}{4\pi \varepsilon_{0}c^{2}R\left(\theta ,z\right)}cos\theta$$
where θ is the emission angle of the THz wave, and $\Phi \left(\omega_{THz},\theta ,z\right)$ and $R\left(\theta ,z\right)$ are the phase and propagation distance of the THz wave, respectively. In this case, the total THz wave emission from the dipole array is the coherent superposition of THz emission from each dipole located at different position along the plasma:(2)$$E_{THz}\left(\omega_{THz},\theta ,L\right)\propto \int_{0}^{L}dE_{dp}\left(\omega_{THz},\theta ,z\right)$$

Thus, the THz emission pattern can be obtained through this integral. Figure 1b shows the THz emission patterns from plasma filaments of different lengths but with the same total emission energy, which are the superposition of calculation results at different frequencies from 0.1 to 3.5 THz with a step size of 0.01 THz. Both the forward and the backward THz emission can be observed from the results of different plasma lengths in Figure 1b. The THz pulses emitted from local dipoles created at different times and positions along the plasma filament have different phases. Owing to the phase differences, the interferences between all the emitted THz pulses in the far field are constructive in the forward direction but destructive in the backward direction. Therefore, the amplitude of the backward THz emission is approximately two or three orders of magnitude lower than that of the forward emission, as shown in Figure 1b. The logarithmic scale in Figure 1b is used to emphasize the backward emitted THz wave, since its amplitude is approximately two or three orders of magnitude smaller than that of the forward emission. Besides, when the plasma length increases, the ratio of backward to forward THz emissions decreases. For a long plasma filament, the THz emission pattern shows a conical distribution, and almost all of the THz energy emitted from the plasma is concentrated in the forward direction, which agrees with the theoretical and experimental results of previous works [13,14]. Investigations on the backward THz emission needs precise alignment of the entire optical system owing to its much weaker amplitude.

In order to detect and investigate the backward THz emission, we designed the experimental setup shown in Figure 1c. In the experiment, the laser beam is delivered by a commercial femtosecond Ti: sapphire amplified laser (Spectra-Physics Hurricane i, Milpitas, CA, USA) with an 800-nm central wavelength, a 600-μJ pulse energy, a 90-fs pulse duration, and a 1-kHz repetition rate. The laser beam is split by a beam splitter (BS) into pump and probe beams. The pump beam has a 7-mm diameter focused by a lens with a 100-mm effective focal length. The HWP placed in front of the lens is used to control the polarization of the pump beam, which optimizes the THz signal, while a 0.1-mm β-barium borate (BBO) crystal is placed behind the lens for the second-harmonic (2ω) generation through frequency-doubling. The relative phase between ω and 2ω is optimized for the THz generation by the means of translating the BBO crystal along the laser axis. The focused ω and 2ω beams pass through the central hole of the parabolic mirror (PM) collinearly, forming a plasma filament around the focus where the THz wave is generated. Figure 1d is a photograph of the plasma filament formed in our experimental configuration. The length of the plasma filament is about 5 mm. The backward emitted THz wave from the plasma filament is collected and collimated by the same parabolic mirror, and then refocused by the second PM. A high resistivity silicon filter is inserted between two PMs to block the light emission from the plasma. Finally, the THz beam and the probe beam are focused collinearly onto a 1-mm thick, <110> cut ZnTe crystal for EOS of the THz waveform.

## 3. Results and Discussions

By the means of EOS, we obtained a typical waveform of the backward THz emission when the pump pulse energy was 480 μJ, as shown in Figure 2a. It is noteworthy that, when the BBO crystal was removed from our experimental configuration, no THz signal could be detected in the backward direction, excluding the possibility of the backward THz signal generated by single-color excitation. The corresponding Fourier transform spectrum of the THz waveform is shown in Figure 2b. As Figure 2b shows, the bandwidth of the backward THz spectrum was ~0.7 THz (full width at half maximum), and the central frequency of the backward THz spectrum was ~0.5 THz, both of which are obviously lower than those of a typical forward THz spectrum [22].Owing to the phase differences between THz pulses emitted from local dipoles created at different times and positions along the plasma filament, the interference of all the emitted THz pulses in the backward direction is destructive. In this case, the spectrum of the backward THz emission was modulated by a frequency-dependent factor originated from the destructive interference. High frequency components were depleted by the destructive interference, reducing the bandwidth and central frequency of the THz spectrum. Therefore, only low frequency components with a narrow bandwidth can be observed in the THz spectrum in Figure 2b. The spectrum of backward THz emissions obtained in our experiment was essentially consistent with the spectra obtained from a micro-plasma [21] and the theoretical prediction [20] in a previous works both of which exhibited a lower central frequency and narrower bandwidth compared with the forward counterpart. 

As demonstrated in a previous work [5], THz electric fields in the forward direction generated by two-color laser fields follow the relationship: $E_{THz}\propto I_{\omega }I_{2\omega }^{0.5}$, where $I_{\omega }$ and $I_{2\omega }$ are the pulse energies of fundamental and second harmonics, respectively. In order to test the dependence of THz amplitude on the laser pulse energy, we measured the total laser pulse energies before the BBO crystal, and recorded the corresponding peak THz amplitudes. The experimental results on the THz wave amplitude as a function of total laser pulse energy, as shown by the black dots in Figure 3a, can be fitted well by function $E_{THz}\propto aI\left(I-bI^{2}\right)$, where I is the total pulse energy, a and b are constants [23]. This function is derived from $E_{THz}\propto I_{\omega }I_{2\omega }^{0.5}$ by replacing $I_{\omega }$ and $I_{2\omega }$ with $I-bI^{2}$ and $bI^{2}$, respectively. The dependence of THz amplitude on the laser pulse energy is the same for backward and forward THz waves in principle. Therefore, we claim that the underlying physics of the THz generation process of the backward THz wave is essentially the same as that of the forward THz wave. Moreover, when the laser pulse energy increases, the THz waveform experiences a shift toward the negative direction in the time domain, as shown in Figure 3b, which corresponds to the decrease of the optical path. The peak shift can be ascribed to the focus shift caused by the nonlinear focusing effect. When the laser pulse energy increases, the laser focus moves ahead of the geometric focus of the lens. In this case, the time-sliced evolution equation for the beam radius $w\left(z\right)$ can be expressed as [24]:(3)$$\ddot{w}\left(z\right)=\frac{4}{k^{2}w^{3}}\left(1-\frac{P_{l}}{P_{cr}}\right)+\frac{4\sqrt{K}\sigma_{K}I_{0}^{K}\tau }{{\left(K+1\right)}^{2}kL_{p}w_{0}}{\left(\frac{w_{0}}{w}\right)}^{2K+1}$$

The first term in the equation represents the contribution of self-focusing effect, while the second term describes the defocusing effect of the ionized electrons. $P_{l}$ and $P_{cr}$ are the laser power and the critical power of self-focusing effect, respectively. $P_{l}$ can be estimated by $P_{l}=I/\tau$, where I and τ are the energy and duration of laser pulses, respectively. In atmosphere, for an 800-nm laser beam, $P_{cr}=3.2\;\mathrm{GW}$. k is the wavenumber of the laser pulse. The initial beam radius and intensity before the lens are defined as $w_{0}$ and $I_{0}$, respectively. K (K=10 for the multiphoton ionization of nitrogen in air) and $\sigma_{K}$ are the multiphoton ionization order and cross-section in air, respectively. $L_{p}=5\;\mathrm{mm}$ represents the plasma length. Taking into account the initial conditions $w\left(0\right)=w_{0}=3.5\;\mathrm{mm}$ and $\dot{w}\left(0\right)=-w_{0}/F_{g}$, where $F_{g}=100\;\mathrm{mm}$ is the geometric focal length of the lens, the nonlinear focal length can be obtained by $F_{nl}=z(\dot{w}=0)$. Therefore, the laser focus shift can be calculated as $∆F=F_{nl}-F_{g}$. The laser focus shift leads to the decrease of the optical paths of both the pump beam and the backward THz beam: $∆L=n_{800}∆F+n_{THz}∆F\approx 2∆F$. $n_{800}$ and $n_{THz}$ are the refraction index of the 800-nm laser beam and the THz wave in air, which can be approximated as 1. The gray dot-dash lines plotted in Figure 3b represent the calculated variations of the optical path as a function of the laser pulse energy, which are fitted well with the positions of the first and the second peaks of the THz waveforms. On the other hand, the peak shift can be seen as evidence of the backward emitted THz wave, since the forward emitted THz signal is not sensitive to the variation of the plasma position. When the plasma moves along the laser axis, the optical path change of the pump beam is opposite to that of the forward THz beam. Hence, their optical path changes be cancelled out by each other: $∆L=n_{800}∆d-n_{THz}∆d\approx 0$, and consequently, the peak position of the forward THz signal does not change. On the contrary, the backward THz signal carries the information of the plasma position, which can be used as a temporal reference in the applications of THz imaging and remote sensing [25,26,27].

In addition, the dependence of the backward THz waveform on the laser pulse chirp was investigated. The chirp of the laser pulse was tuned by adjusting the compression grating in the laser system. The backward THz waveforms as a function of the laser chirp are shown in Figure 4a. When the laser chirp was tuned from positive to negative, the amplitude of the backward THz wave first increased and then decreased, reaching its maximum at a positive chirp near the transform-limit pulse condition, as indicated by the fourth curve from the bottom in Figure 4a. This result is consistent with the dependences of the THz energy on the chirped pulse duration reported in previous works [28,29], which showed that the highest THz energy was achieved using positively chirped laser pulses. Moreover, the laser pulse duration increased with the absolute value of laser chirp, and, consequently, the laser peak power decreased with it, resulting in the temporal shift of the THz waveform caused by the nonlinear-focusing effect, as shown in Figure 4a. The temporal shift of the THz waveform can be roughly fitted by the curve derived from Equation (3) by changing the pulse duration (τ) at a fixed pulse energy of 430 μJ, as shown by the black dashed line in Figure 4a. Bedsides, in order to show the dependence of the THz spectrum on the laser chirp, the Fourier transform spectra of the THz waveforms are illustrated in Figure 4b. When the laser pulse duration increased with the absolute value of laser chirp, the bandwidth and central frequency of the backward THz spectrum decreased, as shown in Figure 4b.

## 4. Conclusions

In conclusion, we theoretically and experimentally investigated the backward THz wave emission from a two-color laser field-induced plasma filament. Theoretically, the linear dipole array model indicates that THz waves can be emitted in both the forward and backward directions from plasma filaments of different lengths. However, the proportion of the backward emitted THz wave decreases with the length of the plasma filament. Experimentally, we obtained the typical waveform and spectrum of the backward THz radiation. The major frequency components of the backward THz spectrum are below 1 THz. Our investigation on the dependence of the peak THz amplitude on the laser pulse energy implies that the THz generation processes of the forward and backward THz waves are essentially the same. Besides, the temporal shift of the backward THz waveform as the laser pulse energy changes can be well interpreted by the nonlinear focusing effect. The sensitivity to the plasma position of the backward THz wave may have potential applications in the fields of THz imaging and remote sensing. Our work could contribute to the further understanding of THz emission process, and may help to develop potential applications of THz sources based on two-color laser-induce plasma filaments.

## Figures and Tables

**Figure 1 sensors-23-04630-f001:**
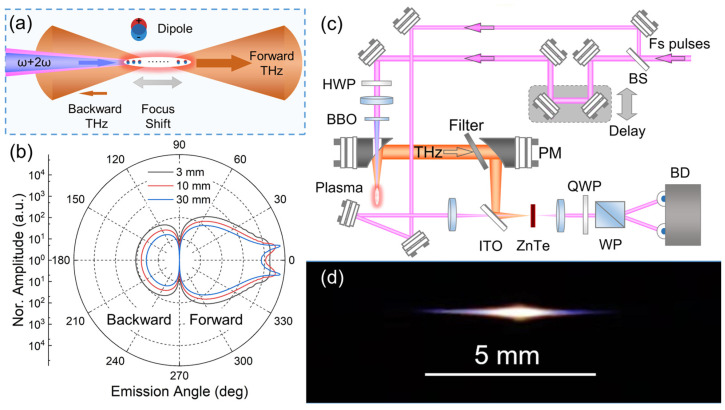
(**a**) The schematic diagram of forward and backward THz emissions from a two-color laser-induced plasma filament. (**b**) Simulation results on THz emission patterns from plasma filaments with different lengths. A logarithmic scale is used for clarifying the relative strength of the backward THz radiation. (**c**) The experimental setup. BS: beam splitter; HWP: half-wave plate; PM: parabolic mirror; QWP: quarter-wave plate; WP: Wollaston prism; BD: balanced detector; ITO: indium tin oxide coated glass plate. Filter: a high-resistivity silicon wafer used to block the light emission from the plasma. (**d**) A photograph of the plasma filament.

**Figure 2 sensors-23-04630-f002:**
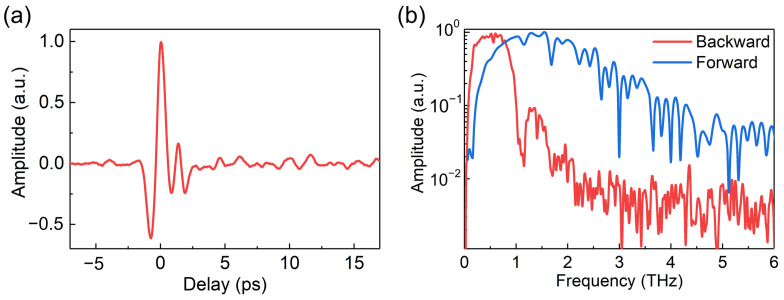
(**a**) A typical waveform of the backward THz emission. (**b**) The spectra of the backward (in red) and forward (in blue) THz emission for comparison.

**Figure 3 sensors-23-04630-f003:**
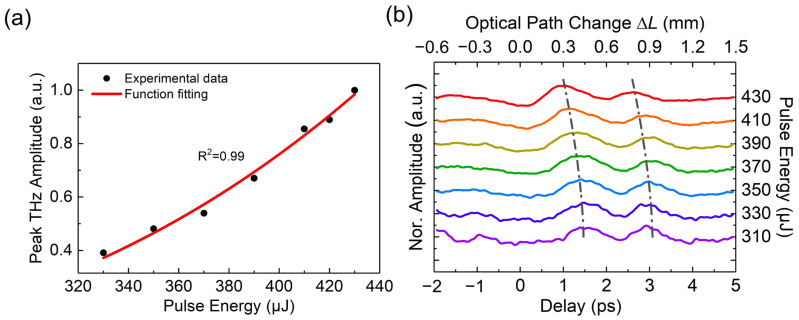
(**a**) The dependence of peak THz amplitude on the total laser pulse energy. (**b**) Solid line: normalized THz waveforms generated by laser pulses with different energies; dot-dashed line: twice the focus shift calculated by the self-focusing formula.

**Figure 4 sensors-23-04630-f004:**
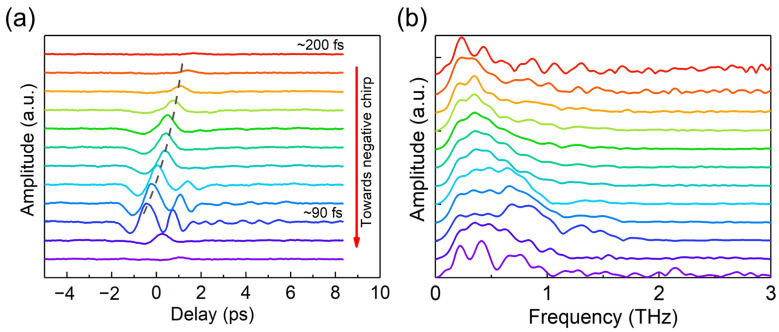
(**a**) THz waveform as a function of the laser pulse chirp. The pulse duration tuned by the laser pulse chirp varied from ~90 fs to ~200 fs. The pulse duration of the transform-limited pulse condition was ~90 fs, corresponding to the THz waveform shown by the dark blue curve. (**b**) The corresponding Fourier transform THz spectrum as a function of the laser pulse chirp. The colors of the curves correspond to those in (**a**).

## Data Availability

The data presented in this study are available on request from the corresponding author. The data are not publicly available due to privacy restrictions.

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
