# Peer review of "Backward THz Emission from Two-Color Laser Field-Induced Air Plasma Filament"

_sensors, 2023, doi:10.3390/s23104630_

Round 1

Reviewer 1 Report

They showed the backward THz emission from two-color laser field-induced air plasma filament. It is an interesting result. I recommend this paper. 

1. The superscript "2" indicating X-C Zhang's affiliation needs to be edited.

Reviewer 2 Report

The manuscript presented by Chen et al. deals with the backward THz emission from two-color laser filamentation. The authors clearly address the progress of backward THz emission from fs laser induced air plasmas. By both theoretical and experimental investigation, the property and mechanism of the backward THz emission were reported. The authors also compared the backward emission with the forward emission. The results may be useful for the community.

I believe that the submitted manuscript is remarkable, well-prepared, and contains very interesting results that deserve to be published in Sensors. I therefore recommend to consider it for publication.

The minor comments I have are:

1.       The authors show the nice simulation results in Fig. 1b. The physical mechanism why the backward emission is two orders’ less intense than the forward emission should be discussed.

2.       The authors present the laser pulse chirp effect on the backward THz emission. (1) In fact, the chirp effect on THz generation from two-color filamentation has been reported before, for example APL 95, 131108 and APL 113, 241103. These works should be mentioned. (2) Better to indicate the transform-limited pulse condition in Fig. 4.

3.       Should the “moment” at line 67 on page 2 be “momentum”?

4.       ITOin Fig. 1c should be indicated in the caption.

Reviewer 3 Report

In this paper, authors theoretically and experimentally investigate the backward THz wave radiation from two-color laser field-induced plasma filament. This work can help to further understand the THz emission process from two-color laser-induced plasma filament.  I recommend acceptance with only a few minor suggestions below:

1. why very few investigations on the backward THz emission have been carried out up to now?

2. peak THz amplitudes in Fig 2(a) and Fig 3(a)  are not matched?

3. please mark the values of the laser pulse chirp in Fig. 4.

Reviewer 4 Report

The paper 'Backward THz emission from two-color laser field-induced air plasma filament' presents the investigation of the backward THz radiation from two color laser field-induced plasma filament. It is indicated in Fig.3(b) of this paper that the peak shift of the THz temporal wavefrom follows well with the focal shift induced by self focusing of filamentation. In this sense, the backward THz signal carries the information of the plasma position. This could have potential applications of THz imaging and remote sensing. This manuscript was interesting to read and was organized nicely. I would recommend this for publication. Just a few questions for consideration.

The authors investigate the focal shift of the 800-nm fundamental pulse, while the THz radiation is via two-color scheme. In this manner, the onset of the focal posiction for second harmonic pulse can be changed by the self-focusing effect of fundamental beam. How will the focal shift of the second harmonic beam affect the peak position of the THz temporal wavefrom? Another question is that the peak power of a pulse can be modified as the chirp was adjusted. Is it possible that the peak shift in Fig.4 also fit with the optical path as a function of the laser peak power as Fig. 3(b)?

Reviewer 5 Report

The proposed paper is well referenced and written. It is accessible even for the non specialist readers. The results you obtained are interesting, but the article suffers from a lack of rigourness and the most important thing to me is that it brings nothing really new to the scientific community concerned by THz generation and their inherent issues, regardless the technique used (photoconductive antennas, optical rectification...).

To illustrate my cruel point of view (since I know how hard it is to deal plasma generation especially when the generated signal id 2 orders of magnitude smaller than frontward generated pulse)...

In this paper, you deal with backward generation of THz pulse from two-color  laser induced plasma, wich is a quite original subject I admit, and a promising technique to adress apllications. In this paper, you specially focused on:

1) a simple model  based on linearly distributed dipoles to simulate the THz emission patern and spectrum,

2) the evolution of the backward radiated THz pulse Vs the laser energy

3) the evolution of the backward radiated THz pulse Vs laser chirp at given energy

Considering the point 1), you never experimentally measured the THz emission pattern to validate your model, and you gave approximative explanation on the shape of the radiated spectrum. For this latter point, I am not agree with you when you say (line 122) that the spectrum is centered at low frequencies (by the way, longer wavelength means lower frequency, contrarily to what you wrote line 124) due to radiation pattern. Indeed, according to diffraction theory, the longer the wavelength (lower the frequency), the more diverging the radiation. In turn, the parabolic mirror acts as a low-cut filter, not a low-pass filter. The radiation pattern cannot explain the low and narrow bandwidth. The real explanation of such spectrum  is well-known as "phase-matching condition" and it is what you call "interferences"... It is true, the phase-matching-condition is explained by interferences between all the THz pulses radiated by each dipole created at different time and space, will act either positively (frontward THz generation) or negatively (backward THz generation) by reducing the bandwidth: sinc shape envelop modifies the spectrum as exhibited in figure 2.b.

It would have been also useful to measure simultaneously the frontward THz pulse to be sure that the sinc envelop observed is not due to the phase condition in the EO detection system that uses a quite thick ZnTe crystal...

Considering the point 2), you should give the values of the parameters occuring in eq(3). Experimentally it could be useful to measure precisely the shape of the plasma using a camera to validate it lengthens with laser energy. Nevertheless, this part is the most interesting of the paper and it deserves to be more detailed.

Considering the point 3), it is known that chirping the laser pulse modifies its duration leading in turn to less or more optical to THz efficiency. At that point it would have interesting to measure the pulse duration Vs the chirp and see if the spectrum depends on the chirp.

I won't go further in my reviewing since I trully beleive that your studies of the backward THz generation deserve more precise explanation, theoretical developpments, and additional measurements for validation. Unless you do that, your paper has poor interest for community.

Dear authors beleive in my kindly regards.

Reviewer 6 Report

This work by Chen et al. reports investigations on the THz emission from two-color laser plasma filaments in the backward direction. The authors present data on the THz emission (time-domain signal, spectrum) of the backward emission and its dependence on pulse energy and laser pulse chirp.

I suggest the authors make a few minor revisions prior to publication.

1)      Generation of THz radiation from two-color plasma filaments: The authors write to generate the plasma filament the laser pulse was guided through a half-wave plate, a lens, and a BBO. The authors state that the half-wave plate was used to control the polarization of the pump beam for optimizing the THz signal. Normally this procedure will result in elliptical polarization of the emitted THz pulse. Could this have an influence on the relative efficiencies of forward- and backward-generated THz wave?

2)      The backward emission shown in Fig. 1 (b) does not show a conical distribution, well-known from the forward emission. Did the authors perhaps measure the beam profile of the emission in the backward direction? It could be interesting in this context.

3)      On p. 3 (ll. 119-131) the authors describe the spectrum of the THz wave in the backward direction. The authors explain the lower center frequency and reduced bandwidth. The explanation given is rather hard to follow for the reader and I would kindly ask the authors to rework this important section, so it becomes clearer. E. g., I do not completely understand what the authors mean in ll. 128/129 “THz waves (…) see relatively shorter equivalent plasma length”.
On a smaller note, in line 124 the authors write “longer wavelength (higher frequency)”. I assume they intended to write ‘lower’ frequency?

4)      The dependence of pulse chirp is interesting and underlines the plausible explanation given by the authors of the change in plasma position. Here, the chirp will also affect the SHG efficiency, which in turn has an influence on the THz conversion efficiency as stated in ll. 136. A smaller note: in ll. 190 the authors write “the laser power decreases with it”. The laser power normally would not decrease by changing the compressor grating. Perhaps the authors mean the peak power?

5)      In ll. 198 the authors state that “lateral THz emission perpendicular to the plasma filament can also be detected in an independent experiment”. I would kindly ask to add the appropriate reference.

Round 2

Reviewer 5 Report

Dear authors,

thank you for providing precision in your manuscript to adress my remarks. But in the revised version I cannot see all the modifications you claim. As exemple I do not see the added figures (R3, R4 and R5).

Please find as attachment my comments about the revised papaer.

Best regards.

Round 3

Reviewer 5 Report

Dear authors,

Thank you for having taken time to answer to all my critics. You did it professionnaly and gently. I will also take into account your advises and remarks considering the performance of our two-color based THz-TDS system.

Best regards.